# New Weed Hosts for Tomato Brown Rugose Fruit Virus in Wild Mediterranean Vegetation

**DOI:** 10.3390/plants11172287

**Published:** 2022-09-01

**Authors:** Nida’ M. Salem, Motasem Abumuslem, Massimo Turina, Nezar Samarah, Abdullah Sulaiman, Barakat Abu-Irmaileh, Yousra Ata

**Affiliations:** 1Department of Plant Protection, School of Agriculture, The University of Jordan, Amman 11942, Jordan; 2Institute for Sustainable Plant Protection-National Research Council (IPSP-CNR), Strada Delle Cacce 73, 10135 Torino, Italy; 3Department of Plant Production, Jordan University of Science and Technology, Irbid 22110, Jordan

**Keywords:** ToBRFV, alternate host, *Solanum nigrum*, *Malva parviflora*, seed transmission

## Abstract

Tomato brown rugose fruit virus (ToBRFV; genus, *Tobamovirus*, family, Virgaviridae) was first reported in 2015 infecting tomatoes grown under protected cropping in the Jordan Valley. Since then, ToBRFV has been detected in tomatoes grown in both protected and open fields across Jordan. The increased incidence of ToBRFV prompted this investigation of the potential role of natural weed hosts in the dissemination of ToBRFV. A survey was conducted in the Jordan Valley and highlands to determine possible reservoir hosts of ToBRFV in fields and greenhouse complexes in which tomatoes were grown. Detection of ToBRFV infection was made by double-antibody sandwich enzyme-linked immunosorbent assay (DAS-ELISA) and further confirmation by reverse-transcription polymerase chain reaction (RT-PCR), followed by DNA cloning and sequencing, and bioassays. Thirty weed species belonging to twenty-six genera from sixteen families were tested. Twelve species belonging to eight families were infected of which ten species are newly reported hosts for ToBRFV. Seed transmission of ToBRFV in *Solanum nigrum* was confirmed in a grow-out experiment. To our knowledge, this is the first report of the natural occurrence of ToBRFV on weed hosts. Identification of natural reservoirs of ToBRFV can help to develop management practices focused on weed plant species to prevent ToBRFV transmission. The extent to which ToBRFV survives in diverse alternate weed host species outside tomato growing seasons in different world regions requires further research in order to establish the risk associated with the possible contribution of weeds as a reservoir for primary infections in tomato crops.

## 1. Introduction

Tomato brown rugose fruit virus (ToBRFV) belongs to the family Virgaviridae, genus *Tobamovirus* and is one of the most devastating plant viruses that cause severe crop losses and threaten tomato production worldwide. ToBRFV was initially described in greenhouse-grown tomatoes in the Jordan Valley in 2015 [1]. Since 2016, this virus has caused serious problems in the production of tomato and pepper crops worldwide [2,3,4,5,6,7,8,9,10,11,12,13,14,15,16,17,18]. To date, the disease has been reported in more than 35 countries across the continents according to the European and Mediterranean Plant Protection Organization EPPO alert list [17].

ToBRFV infection reduces yield and fruit quality as well as plant vigor. It is seed-borne in tomato and eventually can affect the tomato seed trade by preventing the export of tomato seeds produced from areas where ToBRFV is present to areas that are ToBRFV free [19,20,21,22]. Typical symptoms of the disease include mainly leaf mosaic, leaf narrowing, and fruit discoloration, and they may vary depending on the cultivar, stage of infection, and weather conditions.

Given the nature of ToBRFV’s seed transmissibility and stability, the various disease-management measures that have been used so far have achieved very limited effectiveness to control plant infection with the virus [23,24]. Among those measures, the management of uncultivated weeds has not yet been well investigated.

In nature, ToBRFV has been reported only in tomato and pepper. The experimental host range includes species of four botanical families (Amaranthaceae, Apocynaceae, Asteraceae, and Solanaceae). ToBRFV has been detected in common weeds following sap-mechanical inoculation of ToBRFV, including; *Chenopdium album*, *C*. *amaranticolor*, *C*. *murale*, *C*. *quinoa*, *Datura stramonium*, *D*. *metel*, *Emilia sonchifolia*, *Nicotiana benthamiana*, *N*. *clevelandii*, *N*. *debneyi*, *N*. *glutinosa*, *N*. *megalosiphon*, *N*. *occidentalis*, *N*. *rustica*, *N*. *sylvestris*, *N*. *tabacum*, *Physalis angulate*, and *Solanum nigrum* [1,2,25]. Luria, et al. [2] identified *S*. *nigrum* (black nightshade) and *C*. *murale* as potential reservoirs for ToBRFV, based on artificial infection thorough mechanical inoculation, since these two species are common in tomato fields in the Mediterranean basin.

In general, weeds can be directly detrimental to crops, or indirectly by hosting insect vectors or acting as reservoir hosts for viruses and insects. The role of weeds is important in plant virus epidemiology, particularly in virus spread and overwintering [26,27,28]. Weed reservoirs can be a source of inoculum from the crop and mechanical spread via transmission or clothes or pollinators. Weed remains in the greenhouse and between greenhouses after harvest can retain the virus inoculum and contribute to ongoing virus problems for the next growing season. Weeds species have been reported as a reservoir for tobamoviruses such as cucumber green mottle mosaic virus (CGMMV), where *Amaranthus graecizans*, *A*. *muricatus*, *Chrozophora tinctoria*, *Ecballium elatrium*, *Molucella laevis*, and *Withania somnifera* were found to be naturally infected by CGMMV [29].

Weeds pose serious issues, especially to small and subsistent farmers in Jordan. Weed management in fields grown for vegetable productions is not properly executed by most farmers [30]. Major weeds vary according to farming systems. Mostly annual weeds are dominant [30]. Weed density in protected tomato fields is supposedly less than in open field systems [30]. However, if weeds were allowed to grow without proper management, they shed their propagules and pose serious weed problems during the following seasons [30]. In addition to the weeds that were tested for the presence of the virus (Table 1), parasitic weeds, especially the branched broomrape *Orobanche ramosa* and *O. aegyptiaca* are common in many vegetable growing fields including tomatoes, in both the Jordan Valley and the uplands [30]. Other problematic weeds in certain areas include *Amaranthus graecizans*, *Chenopodium vulvaria*, and many types of grass such as *Cynodon dactylon* [30].

ToBRFV may infect agricultural weed species that may allow the persistence of the virus in periods with no tomato production. However, the role of weeds in the disease epidemiology of ToBRFV is unknown. In general, the detection of weeds that act as a natural reservoir for the disease is difficult since many different weeds can be infected with viruses without showing symptoms, as is becoming evident from transcriptomic NGS analysis of plant species in natural settings [31,32,33]. In this study, the presence of alternate weed hosts and seed transmissibility in *S*. *nigrum* were investigated for the first time. This paper presents the results of a survey to collect information on the occurrence of ToBRFV in weeds in tomato-grown greenhouses and open fields. This information on the natural host range of ToBRFV, including potential alternate hosts that could serve as virus reservoirs, is useful for a better understanding of the disease epidemiology and in developing an integrated management strategy for reducing the impact of ToBRFV.

## 2. Results

### 2.1. Incidence of ToBRFV in Weeds

A wide range of weeds was found in tomato fields. A total of 258 samples representing 30 plant species corresponding to 16 families were tested for the presence of ToBRFV by DAS-ELISA (Table 1). Collected weed plants were almost all asymptomatic except for a few exceptions that showed yellowing, stunting, or mosaic, and mottling (Figure 1). ToBRFV was detected by DAS-ELISA in 114 samples, corresponding to 12 plant species belonging to 8 families: *Amaranthus retroflexus*, *Beta vulgaris* subsp. *Maritima*, and *Chenopodium murale* (Amaranthaceae); *Conyza Canadensis* and *Taraxacum officinale* (Asteraceae); *Malva parviflora* (Malvaceae); *Oxalis corniculata* (Oxalidaceae); *Portulaca oleracea* (Portulaceae); *Veronica syriaca* (Scrophulariaceae); *Solanum elaeagnifolium* and *S*. *nigrum* (Solanaceae); and *Corchorus olitorius* (Tiliaceae). Infection was confirmed by mechanical inoculation to *N*. *tabacum*, *D*. *metel*, and *D*. *stramonium*. Symptoms of the indicator plants were similar to those observed earlier in a previous study [1]. Most infected plants belonged to species in the Solanaceae, Amaranthaceae, and Malvaceae, followed by the Scrophulariaceae, Tiliaceae, Asteraceae, Portulacaceae, and Oxalidaceae (Table 1). Plant species where more than 50% of the collected samples were infected, include *A*. *retroflexus*, *C*. *canadensis*, *T*. *officinale*, *C*. *murale*, *M*. *parviflora*, *V*. *syriaca*, and *S*. *nigrum* (Table 1).

ToBRFV could not be detected by DAS-ELISA in plants of 18 weed species from 8 families. These species were *Amaranthus viridis* (eight plants tested), *Ammi majus* (two), *Anthemis* sp. (one), *Lactuca serriola* (one), *Matricaria aurea* (two), *Senecio jacobaea* (two), *S*. *vernalis* (three), *Sonchus oleraceus* (eight), *Convolvulus arvensis* (fifteen), *Bryonia cretica* (one), *Mercurialis annua* (two), *Melilotus italicus* (three), *Trifolium clusii* (two), *T*. *repens* (two), *Fumaria densiflora* (two), *Anagallis arvensis* (two), *Withania somnifera* (two), and *Urtica urens* (two). The number of plants tested, belonging to these species was low except *C*. *arvensis*.

To verify the ELISA results for the 12 ToBRFV-positive weed species, RT-PCR was conducted. An approximately 872-bp DNA fragment was amplified by RT-PCR with a ToBRFV RdRp primer pair (Appendix A). No DNA fragment was amplified from uninfected (ELISA negative) plants. The DNA fragment amplified from the ToBRFV infected plants was cloned, and its nucleotide sequence was determined. DNA sequence comparisons revealed that the sequence of this DNA fragment was 99% identical to the RdRp partial sequence of ToBRFV-Jo isolate (KT383474) recovered from a tomato from the Jordan Valley (Salem et al., 2016). Calculated pairwise nucleotide identities between the 12 Jordanian isolates from weeds ranged between 98.97 and 100%. When ToBRFV RdRp sequences from the NCBI database were included in the analysis, percentages of nucleotide identity varied between 98.62 and 100%. However, alignment of the 12 nucleotide sequences from weeds indicated that single nucleotide mutations are present in different isolates in different positions of the segment, and they are all synonymous (data not shown).

### 2.2. ToBRFV Transmission from Tomato to Malva parviflora

The susceptibility of *M*. *parviflora* to ToBRFV was confirmed by artificial mechanical inoculation of healthy *M*. *parviflora* seedlings. Although no visual symptoms developed at 30 to 60 dpi, all inoculated plants were ELISA and RT-PCR positive, confirming that this species can be infected by ToBRFV.

### 2.3. Virome Characterization of Symptomatic Solanum nigrum Sampled from a Field

After raw reads were processed, a total of 52,902,748 paired-end reads of 101 bp were obtained, generating 72,268 contigs with de novo assembly using the Trinity program. BLASTn analysis of the assembled contigs revealed the presence of one virus-derived contig: ToBRFV (6383 nt, accession no. OP009342) in the *S*. *nigrum* sample, which represented a nearly full-length genome. ToBRFV was well represented by the number of reads mapping the genome, where the read count was about 37,196. The ToBRFV-related contig shared more than 99.86% nucleotide sequence identity with the ToBRFV Jordanian isolate (KT383474) from tomato.

### 2.4. Seed Transmission of ToBRFV in Solanum nigrum

Results from a grow-out experiment of *S*. *nigrum* seed indicated a seed transmission rate of approximately 1.9% (2/107) from contaminated seeds to healthy seedlings. The infected *S*. *nigrum* seedlings developed mild mosaic symptoms and tested positive for DAS-ELISA. ELISA absorbance value (A405) of the positive seedlings ranged from 0.557–0.813, which was higher than the cut-off value for twofold the negative control (0.206), and also compared with other negative seedlings within the same experiment ranging from 0.149 to 0.202. Mechanical inoculation of the leaf extract from the two infected seedlings to indicator plants was demonstrated in all the tested indicator hosts and further confirmed by RT-PCR. In tomato inoculated plants the infection was confirmed only based on RT-PCR, and due to accidental *Tuta absoluta* infestation symptoms were not evaluated. All seedlings that originated from *S*. *nigrum* healthy seeds were virus-free based on the DAS-ELISA test and RT-PCR, confirming that no accidental contamination occurred in our experiment.

## 3. Discussion

Since the first detection of ToBRFV in Jordan in 2015, it has been consistently detected every year in tomato cropping areas across the country. These continuous outbreaks raised concern about the possible role of weed hosts of ToBRFV in the major tomato growing areas. Results of this study revealed the presence of the virus in 12 wild species, 10 of them are new hosts for ToBRFV whereas *C*. *murale* and *S*. *nigrum* were shown to be mechanically infected by ToBRFV previously [2]. Here, we show that both *C*. *murale* and *S*. *nigrum* are natural hosts of ToBRFV (Table 1). Seven of the identified host species are annuals and five perennials. Both annual and perennial weeds may play an important role in virus epidemiology by forming a “green bridge” when tomatoes are not present and facilitating the seasonal carryover of the virus.

Several approaches in addition to DAS-ELISA were used to prove that weeds were infected with ToBRFV. First, ToBRFV was mechanically transmitted from infected weeds to indicator plants; these mechanically inoculated plants developed distinctive symptoms of ToBRFV infection and were confirmed positive for ToBRFV infection based on DAS-ELISA results. Second, ToBRFV infection was detected by RT-PCR amplification, cloning, and sequencing of a portion of the RdRp gene. The sequence of the RdRp fragment amplified from 12 infected weed species was 99% identical to the sequence of the ToBRFV RdRp fragment amplified from ToBRFV-infected tomato (ToBRFV-Jo) in the Jordan Valley [1]. Minimal differences in sequences are still consistent with a single undifferentiated clonal infection of ToBRFV in Jordan, and minimal sequence alteration can be because of the quasispecies mix of nearly identical genomes in the original samples, or because of the influence of the Taq polymerase error rate during the amplification step. Taken together, these results identify the field-collected weeds as natural ToBRFV hosts. However, a further confirmatory in vivo assay to show that ToBRFV infecting weeds can potentially also infect tomato should be included in future studies.

Although ToBRFV was identified as infecting twelve common weeds, only a few of them showed virus symptoms. However, common weeds, often asymptomatic when infected by the virus, comprise a cryptic reservoir between consecutive growing cycles. The higher genetic diversity of weeds in comparison to crops, the presence of numerous different weed species in an area, which limits contact between identical, susceptible plants, and long-term selection of tolerance or resistance often results in asymptomatic virus infection in weed hosts [34,35].

A significant finding of this study was the identification of *M*. *parviflora* as a new and widespread weed host of ToBRFV in Jordan. *M*. *parviflora* plants were widespread in the Jordan Valley and were found near ToBRFV hot spots. In general, *M*. *parviflora* has previously been reported as a reservoir for several plant viruses such as alfalfa mosaic virus, tomato yellow leaf curl virus (TYLCV) and tomato spotted wilt virus [36,37,38]. Additionally, it can act as a host for whiteflies and thrips which can vector viruses from the surrounding weeds to the crop.

Interestingly, ToBRFV was detected in two weed species of the genus *Solanum*; *S*. *elaeagnifolium* (silverleaf nightshade) and *S*. *nigrum* (black nightshade). *S*. *nigrum* showed obvious virus-like symptoms (specifically, leaf mottling and mosaic) compared to the other collected weeds (Figure 1a). The presence of ToBRFV was associated with symptomatic plants and further confirmed by virome characterization. Deep sequencing and de novo assembly of RNA from symptomatic *S*. *nigrum* indicated the presence of only ToBRFV. The full-length characterization by NGS of the *S. nigrum* isolate did not show any sign of specific adaptation and has only a minimal number of mutations compared to the tomato-infecting ones. Black nightshade is widespread throughout the world, specifically in tropic and sub-tropic regions. It is also a major weed problem in tomato fields in the Jordan Valley and the highlands throughout Jordan. It is a highly competitive weed that lowers crop yield and quality, and it can interfere with the harvest. It is considered a prolific seed producer. The seeds begin germinating when the weather starts warming up and can continue to germinate throughout the growing season. However, the management of this weed in tomato fields is not an easy task. Weed remains should be removed from the field after the final crop harvest, because survivors replenish the seed bank and create a problem in the field in future years. On the other hand, *S*. *elaeagnifolium* infected plants did not show any symptoms and were only found at the borders and between greenhouses in one tomato field surveyed in this study. It is worth mentioning that *S*. *elaeagnifolium* has been spreading in waste areas and some cultivated fields in Jordan but it is not yet as invasive compared to its situation in many countries [39]. However, our study highlights the urgent need for monitoring *S*. *elaeagnifolium* and implementing preventive measures into the current agricultural management strategies.

ToBRFV was found in jute (*Corchorus olitorius*), which is an important bast (phloem) fiber crop that is mainly grown in Southeast Asian countries such as India, Bangladesh, Nepal, China, and Indonesia. However, in Jordan, the tender leaves of jute plants are consumed as green leafy vegetables (mulukhiyah). This plant has been reported to harbor some viruses such as TYLCV, cucumber mosaic virus, and papaya ring spot virus [40,41,42]. The presence of ToBRFV in *C*. *olitorius* is significant as this plant could serve as a reservoir for the virus playing a crucial role in spreading ToBRFV in the tomato fields as well as tomatoes grown in home gardens.

ToBRFV transmission through seed has been reported in tomato [43,44], but with a low rate. Similarly, seed transmission occurs with a low rate (1.9%) from ToBRFV-contaminated *S*. *nigrum* seeds to seedlings, which could initiate disease and drive further dissemination of the disease in the same field and new growth areas. The relatively higher rate of seed transmission obtained in *S*. *nigrum* compared with tomato (0.08%) in our previous work [44] could be attributed to the fact that seeds used in the growing-out test of *S*. *nigrum* were directly extracted from fruits and sown in peat moss media without any washing with sterile distilled water or treatment with disinfectants, to mimic open field conditions.

Overall, this study underlines the necessity to reconsider control measures in tomato growing practices before planting the following growth cycle since weeds might carry a primary inoculum of ToBRFV. Elimination of alternate host plants, and removal of leftover plant debris after harvesting and weed removal, eventually controlled through the burning of weeds play a significant role in an attempt to prevent disease spread [34]. Farmers should also employ regular weeding strategies to avoid weed build-ups that might carry ToBRFV.

## 4. Materials and Methods

### 4.1. Plant Materials

Weeds were sampled from February 2019 to November 2021 in tomato fields of the Jordan Valley and highlands in the main tomato growing areas of Jordan, which showed high ToBRFV disease incidence. The weeds were randomly collected within or surrounding the affected greenhouses regardless of the presence of symptoms. Plants at the borders and between the greenhouse units were also sampled. The sample number of every plant species collected was related to the abundance of the species at the time of collection. The material sampled consisted mostly of the entire plants including leaves and flowers when present to assist in species identification. For virus testing, the youngest fully expanded leaf was used. The plant species of the collected samples were identified according to the botanical classification key [30,45]. Additionally, ripe fruits of *S*. *nigrum* were collected from symptomatic plants for further experiments to assess seed transmission of ToBRFV.

### 4.2. Detection of ToBRFV in Weeds by ELISA, RT-PCR, and Bioassay

Samples were analyzed for ToBRFV infection by double-antibody sandwich enzyme-linked immunosorbent assay (DAS-ELISA) using a ToBRFV commercial kit (LOEWE Biochemia, Germany) following the manufacturer’s instructions. Plant extracts were prepared by grinding leaf tissue at a ratio of 1:20 (wt/vol) in a commercial grinding buffer from Biroeba (Bioreba, AG, Reinach, Switzerland). The absorbance values were read on a microplate ELISA-reader (Biochrom Asys Expert Plus Microplate Reader, Biochrom, UK) at OD405 nm. Samples with values higher than twice the average ELISA values of samples from healthy plants were considered positive.

Reverse transcription polymerase chain reaction (RT-PCR) was carried out with weed samples that tested positive for ToBRFV infection by DAS-ELISA. The identity of ToBRFV was confirmed by RT-PCR using the primer set ToBRFV-F (5′-CATATCTCTCGACACCAGTAAAAGGACCCG-3′) and ToBRFV-R (5′-TCCGAGTATAGGAAGACTCTGGTTGGTC-3′) targeting a fragment of the RNA dependent RNA polymerase (RdRp) designed on ToBRFV genome (KT383474) [1]. Total RNA was extracted from leaves of ELISA-positive as well as negative plants of the same species, using an SV-Total RNA Extraction kit (Promega, Madison, WI, USA) following the manufacturer’s instructions. RT-PCR products (872 bp) of 12 samples were purified and ligated into pGEM T-Easy Vector (Promega, Madison, WI, USA), and two clones for each PCR product were bidirectionally sequenced (Macrogen, Seoul, Korea), and results were deposited in NCBI GenBank (OP009331-OP009341, OP066373). The nucleotide sequences were compared with sequences available from the NCBI database using BLASTn program in GenBank (https://blast.ncbi.nlm.nih.gov/Blast.cgi), accessed on 27 July 2022. Sequences from different hosts and origins were aligned with ClustalX [46], then visualized and analyzed with the Molecular Evolutionary Genetics Analysis Version X (MEGA X) software [47].

Additionally, positive samples for ToBRFV by ELISA and RT-PCR were mechanically inoculated onto *N*. *tabacum* cv. White Burley, *D*. *metel*, and *D*. *stramonium* to confirm the identity and infectivity of the virus present in the weed. The inoculums were prepared by grinding the leaves of infected weed species (ToBRFV-ELISA and RT-PCR positive) in cold potassium phosphate buffer (0.01 M, pH 7 containing 0.001 M DIECA and 0.001 M cysteine) using chilled mortar and pestle. Indicator plants were dusted with 600-mesh carborundum and the extract was gently rubbed on the leaves. Three plants of each indicator species for each positive weed species (ToBRFV-ELISA and RT-PCR positive) were used. Control plants were mock inoculated to exclude contamination from buffer and watering practices. Plants were kept in the greenhouse at 24 ± 2 °C and were continuously examined for local and systemic symptoms for up to 21 days. All indicator plants were tested with DAS-ELISA for back indexing.

### 4.3. Ability of ToBRFV Transmission from Tomato to Malva parviflora

Following the analysis of collected weed species by ELISA, ToBRFV was found at a high frequency (16/23) in *M*. *parviflora* collected samples. To verify the susceptibility of *M*. *parviflora* to ToBRFV, mechanical inoculation of healthy seedlings of *M*. *parviflora* was performed. ToBRFV was maintained on systemically infected tomato plants cv. Hybrid tomato super red and the inoculated tomato plants served for further sap-mechanical inoculation of test plants. Mature seeds of *M*. *parviflora* were collected from plants grown in an aseptic greenhouse at the University of Jordan that had tested negative for ToBRFV based on the ELISA test. They were sown in peat moss, and two weeks later seedlings were transplanted in pots containing sterilized soil. Ten malva seedlings were mechanically inoculated by ToBRFV as mentioned above using ToBRFV tomato infected leaves and kept in a plant growth chamber with a 14 h light/10 h dark cycle at 24 ± 2 °C for two months. Virus infection was evaluated by observation of symptoms and virus detection by ELISA and RT-PCR at 30 days post inoculation (30 dpi) and 60 dpi; the experiment was repeated twice.

### 4.4. Virome Characterization of Symptomatic Field Solanum nigrum

Total RNA was extracted from leaves of *S*. *nigrum* showing virus-like symptoms collected from a tomato field (30°21′11.3′′ N 35°37′10.3′′ E) on 16 October 2021, with innuPREP Plant RNA Kit (Analytik Jena AG, Jena, Germany) according to the manufacturer’s specifications. The presence of ToBRFV was confirmed by DAS-ELISA and RT-PCR using the same protocols mentioned before. RNA integrity was checked using Agilent Bioanalyzer 2100 system (Agilent Technologies, Santa Clara, CA, USA). The cDNA library was constructed using a TruSeq Stranded Total RNA with Ribo-Zero Plant Kit (Illumina Inc., CA, USA) and sequenced by Illumina a NovaSeq600 platform (Macrogen Inc., Seoul, Korea). Trimmomatic program was used to remove adapter sequences and bases with base quality lower than three from the ends [48]. De novo RNA-Seq transcript assembly was conducted using the Trinity program [49]. All de novo assembled contigs were blasted against public databases with BLASTn and BLASTx of NCBI.

### 4.5. Evaluation of Seed Transmission of ToBRFV in Naturally Infected Solanum nigrum

Ripe fruits of *S*. *nigrum* were collected from symptomatic plants grown in the tomato field mentioned above. ToBRFV infection was confirmed in fruits by DAS-ELISA. A grow-out experiment was conducted to assess the vertical transmission (seeds to progeny seedlings) of ToBRFV in *S*. *nigrum*. Seeds were extracted directly from the fruits without any treatments and sown in peat moss on 24 October 2021. One month later, seedlings were transplanted individually into a small disposable clean plastic pot (9 × 9 × 8 cm) (350 mL) in a substrate mixture of peat moss and perlite at a ratio of 2:1 (vol/vol). One hundred and seven seedlings were grown for two months at the University of Jordan, Amman, Jordan, under controlled conditions of 24 ± 2 °C and 16 h light with stringent sanitation procedures. Plants were observed daily for symptom development. Samples of leaf tissues were collected from each seedling and tested by DAS-ELISA two months after transplanting. Strict sanitation measures were taken to avoid any possibility of contamination during sample collection. Healthy seeds obtained from fruits of *S*. *nigrum* plants collected in a pear cactus field in Madaba (31°35′36.8′′ N 35°51′00.4′′ E) free from vegetable crops were included in the whole experiment as a negative control. To verify the presence of ToBRFV in the DAS-ELISA positive seedlings, two methods were used: RT-PCR and mechanical inoculation of three indicator plant species (*C*. *quinoa*, *D*. *metel*, and *D*. *stramonium*) and tomato cv. Hybrid tomato super red.

## 5. Conclusions

The majority of weed species found to be ToBRFV-infected were very common in the tomato growing areas of the region. This indicates that tomato fields contaminated with these weed species are at risk of viral infections. Further studies are required to determine whether additional weed species from different world regions can be ToBRFV hosts, to understand how long the virus can persist in weed species hosts when the tomato crop is absent, and whether seed transmission occurs from weed seeds other than *S*. *nigrum*. Finally, future work should address the efficacy (and cost vs. benefit) of weed management practices in containing epidemics of ToBRFV in open fields and greenhouses.

## Figures and Tables

**Figure 1 plants-11-02287-f001:**
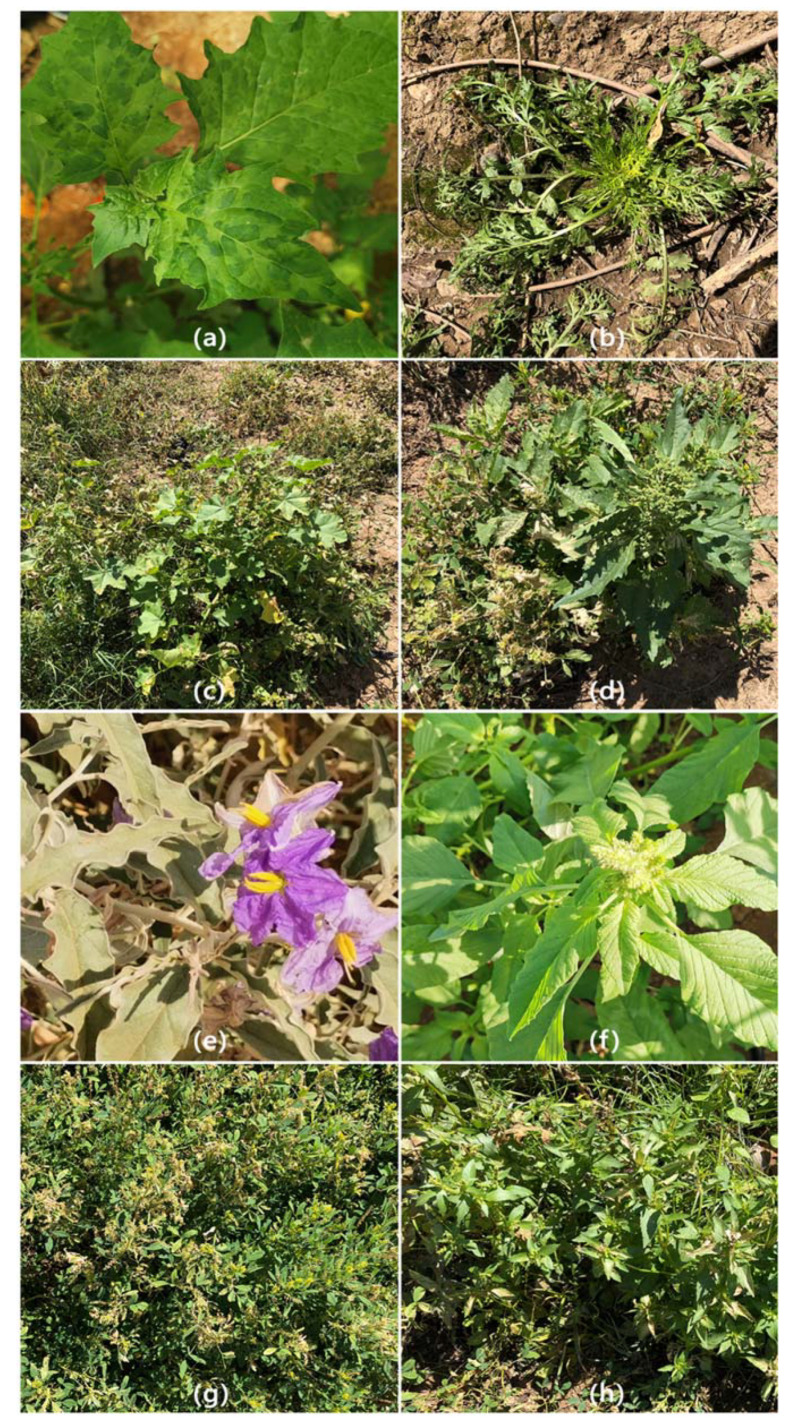
Weeds collected from tomato fields. Weeds were collected from tomato fields showing virus-like symptoms including: (**a**) *Solanum nigrum* with mosaic and mottling in the upper leaves; (**b**) *Senecio jacobaea* with stunting; (**c**) *Malva parviflora* with yellowing of the basal leaves. Weeds were almost asymptomatic as in (**d**) *Chenopodium murale*; (**e**) *Solanum elaeagnifolium*; (**f**) *Amaranthus retroflexus*; (**g**) *Melilotus italicus*; and (**h**) *Corchorus olitorius*.

**Table 1 plants-11-02287-t001:** Plant species collected and results of ELISA testing for tomato brown rugose fruit virus.

Family	Weed Species	Life Cycle	No. of Collected Plants	No. of Plants Positive
Amaranthaceae	***Amaranthus retroflexus*** **	SA	32	18
	*Amaranthus viridis*	P	8	0
	** *Beta vulgaris* ** ** subsp. *Maritima***	B; P	6	2
	*Chenopodium murale*	SA	17	11
Apiaceae	*Ammi majus*	A	2	0
Asteraceae	*Anthemis* sp.	A	1	0
	** *Conyza canadensis* **	WA	2	1
	*Lactuca serriola*	SA; B	1	0
	*Matricaria aurea*	A	2	0
	*Senecio jacobaea*	WA; B	2	0
	*Senecio vernalis*	A	3	0
	*Sonchus oleraceus*	SA/WA	8	0
	** *Taraxacum officinale* **	P	4	2
Convolvulaceae	*Convolvulus arvensis*	P	15	0
Cucurbitaceae	*Bryonia cretica*	P	1	0
Euphorbiaceae	*Mercurialis annua*	A	2	0
Fabaceae (=Leguminosae)	*Melilotus italicus*	A	3	0
	*Trifolium clusii*	A	2	0
	*Trifolium repens*	p	2	0
Malvaceae	** *Malva parviflora* **	A; P	23	16
Oxalidaceae	** *Oxalis corniculata* **	P	1	1
Papaveraceae	*Fumaria densiflora*	A	2	0
Portulacaceae	** *Portulaca oleracea* **	SA	9	2
Primulaceae	*Anagallis arvensis*	A	2	0
Scrophulariaceae	** *Veronica syriaca* **	A	9	6
Solanaceae	** *Solanum elaeagnifolium* **	P	26	7
	*Solanum nigrum*	A	61	46
	*Withania somnifera*	P	2	0
Tiliaceae	** *Corchorus olitorius* **	A	8	2
Urticaceae	*Urtica urens*	A	2	0
Total			258	114

A = annual; SA = summer annual; WA = winter annual; B = biennial; P = perennial. ** Newly identified host species are given in bold. The specimen of each weed included in the table was kept in the weed unit at the School of Agriculture at the University of Jordan.

## Data Availability

There are no data related to this paper that are not presented.

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
