# Peer review of "New Weed Hosts for Tomato Brown Rugose Fruit Virus in Wild Mediterranean Vegetation"

_plants, 2022, doi:10.3390/plants11172287_

Round 1

Reviewer 1 Report

In the submitted manuscript, Salem et al describe determination of ToBRFV host range in wild and native plant species located in or around commercial tomato production. This topic is of vital importance for consideration of management approaches of this devastating pathogen. The authors used primarily ELISA-based techniques that were verified by RT-PCR to identify ToBRFV in various plant species. In addition, the authors examined the presence of ToBRFV more specifically in two plant species which are commonly present in tomato producing areas to determine vertical descent of ToBRFV in these weed species, and to determine the genome sequence of ToBRFV from non-cultivated plants. This manuscript identifies a number of potential weed species that could act as reservoirs of ToBRFV, and helps to identify weed management priorities to help prevent transmission to cultivated tomato crops. While the manuscript is generally well written, there are a number of detracting factors that should be improved. Major issues include a weak and short introduction, and an overly lengthy discussion, a number of grammar and sentence structure issues, poor figure resolution.

Major issues: Ln 24 awkward wording. ln 24: "This finding leads to the inclusion of weed control as part of the ToBRFV 24 management strategy as a cautionary measure." Suggest instead "Identification of natural reservoirs of ToBRFV can help to develop management practices focused on weedy plant species to prevent ToBRFV transmission". The authors have only provide information that can inform management practices, but there is no requirement for anyone to include these measures. The wording in the manuscript is too strong.

Further background in the introduction regarding what weed species are commonly present in tomato production regions would be beneficial. The discussion section is quite long, and lines 226-233 might be better placed in the introduction. Overall the discussion is meandering, and editing to be more concise and focused would be highly beneficial. 

ln 23 "this is the first report on the natural occurrence of ToBRFV in weed species" but the authors cite multiple papers that have identified ToBRFV in weed species. This is contradictory.

What precisely is a grow out experiment? I am not familiar with this term, and it is not well defined. Presumably this is a test for vertical (seed) transmission of ToBRFV, and could be better explained using proper botanical and virology notation.

Figure 1 is of poor quality images and it is not clear what is being displayed. Improved resolution and labelling would improve this figure. 

Figure S1 legend is missing. There is a figure caption, but this is not an appropriate figure legend. Agarose gel percentage should be in the Materials section and not the figure caption.

Minor issues: ln 14 has repetitive use of "increasing"

Author Response

Response to Reviewer 1 Comments

In the submitted manuscript, Salem et al describe determination of ToBRFV host range in wild and native plant species located in or around commercial tomato production. This topic is of vital importance for consideration of management approaches of this devastating pathogen. The authors used primarily ELISA-based techniques that were verified by RT-PCR to identify ToBRFV in various plant species. In addition, the authors examined the presence of ToBRFV more specifically in two plant species which are commonly present in tomato producing areas to determine vertical descent of ToBRFV in these weed species, and to determine the genome sequence of ToBRFV from non-cultivated plants. This manuscript identifies a number of potential weed species that could act as reservoirs of ToBRFV, and helps to identify weed management priorities to help prevent transmission to cultivated tomato crops. While the manuscript is generally well written, there are a number of detracting factors that should be improved. Major issues include a weak and short introduction, and an overly lengthy discussion, a number of grammar and sentence structure issues, poor figure resolution.

Point 1:  Major issues: Ln 24 awkward wording. ln 24: "This finding leads to the inclusion of weed control as part of the ToBRFV management strategy as a cautionary measure." Suggest instead "Identification of natural reservoirs of ToBRFV can help to develop management practices focused on weedy plant species to prevent ToBRFV transmission". The authors have only provide information that can inform management practices, but there is no requirement for anyone to include these measures. The wording in the manuscript is too strong.

Response 1: Yes, we agree with the reviewer. The sentence has been re-written as suggested.

Point 2: Further background in the introduction regarding what weed species are commonly present in tomato production regions would be beneficial. The discussion section is quite long, and lines 226-233 might be better placed in the introduction. Overall the discussion is meandering, and editing to be more concise and focused would be highly beneficial. 

Response 2:

  • One paragraph covering the weeds species that are commonly present in Jordan has been added to the introduction.
  • Lines 226-233 were moved from the discussion and placed in the introduction.
  • The discussion was completely revised and edited and it looks more concise and focused now.

Point 3: ln 23 "this is the first report on the natural occurrence of ToBRFV in weed species" but the authors cite multiple papers that have identified ToBRFV in weed species. This is contradictory.

Response 3: With respect to the author comment, our statement is correct; it refers to natural infection. The other papers that reported C. murale and S. nigrum as hosts, are mainly relied on artificial (mechanical) inoculation of the plant and this was already mentioned in the introduction..

Point 4: What precisely is a grow out experiment? I am not familiar with this term, and it is not well defined. Presumably this is a test for vertical (seed) transmission of ToBRFV, and could be better explained using proper botanical and virology notation.

Response 4: The grow-out experiment was conducted to assess the vertical transmission (seeds to progeny seedlings) of ToBRFV in S. nigrum. This statement is clearly mentioned under the Materials and Methods section 4.5. The term “grow-out test” is commonly used in seed transmission paper, and has been used in text books such as

  1. Subramanya Sastry. Seed-borne Plant Virus Diseases. Springer. 2013. Page 107. (attached)
  2. Ravindra Kumar and Anuja Gupta. Seed-Borne Diseases of Agricultural Crops: Detection, Diagnosis & Management. Springer. 2020. Page 175. (attached)

Point 5: Figure 1 is of poor quality images and it is not clear what is being displayed. Improved resolution and labelling would improve this figure. 

Response 5: The resolution and labelling of Figure 1 have been improved.

Point 6: Figure S1 legend is missing. There is a figure caption, but this is not an appropriate figure legend. Agarose gel percentage should be in the Materials section and not the figure caption.

Response 6: Figure S1 legend is available in the manuscript itself after the conclusion according to the guidelines for authors. However it has been added to the supplementary too as suggested. Agarose gel percentage was removed.

Point 7: Minor issues: ln 14 has repetitive use of "increasing"

Response 7: It has been deleted.

Point 8: The referee suggested that the manuscript should undergo English revisions.

Response 8: The manuscript has been checked and edited by a native-English speaking colleague (Sue. T. Sim from Foundation Plant Services, UC Davis). Acknowledgements section was added to the manuscript.

Reviewer 2 Report

The manuscript of Salem et al.:" New Weed Hosts for Tomato Brown Rugose Fruit Virus in Wild Mediterranean Vegetation" report on a survey conducted on weeds grown in greenhouses of ToBRFV infected tomato plants. A wide variety of weed families was found infected with ToBRFV. The fact that the authors have found ToBRFV infection of the weeds in tomato growing areas emphasizes the importance of the investigated wild plants as reservoirs of the virus between growing seasons.

Remarks to the authors:

1.       Line 68: "few of the collected weed plants showed virus symptoms". The authors refer to Figure 1 but symptoms were apparent only in f-S. nigrum. It would be very important to describe symptoms on the investigated plants. Symptoms may be more apparent if ToBRFV inoculations were conducted as well similar to the inoculation done on Malva parviflora (Line 112).

2.       Line 74: It would be important to describe and show the results with indicator plants. In addition, the authors refer to ref [1] of Salem 2016 but D. metel and D. stramonium were not used as indicator plants in that publication.

3.       Lines 97-110: The authors have compared nucleotide sequence identity only of a fragment of 872 bp and found synonymous mutations (except for S. nigrum, lines117-125). It is very important to confirm that ToBRFV in the investigated wild plants was not modified and the plants indeed could act as reservoirs for ToBRFV infection of tomato plants by using sap from the wild plants not only to infect indicator plants but also to infect tomato plants.   

Author Response

Response to Reviewer 2 Comments

The manuscript of Salem et al.:" New Weed Hosts for Tomato Brown Rugose Fruit Virus in Wild Mediterranean Vegetation" report on a survey conducted on weeds grown in greenhouses of ToBRFV infected tomato plants. A wide variety of weed families was found infected with ToBRFV. The fact that the authors have found ToBRFV infection of the weeds in tomato growing areas emphasizes the importance of the investigated wild plants as reservoirs of the virus between growing seasons.

Point 1: Line 68: "few of the collected weed plants showed virus symptoms". The authors refer to Figure 1 but symptoms were apparent only in f-S. nigrum. It would be very important to describe symptoms on the investigated plants. Symptoms may be more apparent if ToBRFV inoculations were conducted as well similar to the inoculation done on Malva parviflora (Line 112).

Response 1: The symptoms on the plant have been added to Figure 1. We agree with the reviewer that the symptoms may be more apparent if ToBRFV inoculations were conducted to the weed plants (but this would mis-guide their relevance, since what matters is how they appear in field conditions). However, it was not easy to obtain seeds from all weeds that tested in our work. But what we have noticed on the weeds that most of them were asymptomatic and this is in line with several published literatures about weeds as a reservoir for plant viruses.

Point 2: Line 74: It would be important to describe and show the results with indicator plants. In addition, the authors refer to ref [1] of Salem 2016 but D. metel and D. stramonium were not used as indicator plants in that publication.

Response 2: The symptoms observed on the indicator plants were similar to what we reported in our previous article (Salem et al., 2016). This article has a supplementary table (Supplementary Table S1) that shows the response of different indicator plants to mechanical inoculations (attached).

Point 3: Lines 97-110: The authors have compared nucleotide sequence identity only of a fragment of 872 bp and found synonymous mutations (except for S. nigrum, lines117-125). It is very important to confirm that ToBRFV in the investigated wild plants was not modified and the plants indeed could act as reservoirs for ToBRFV infection of tomato plants by using sap from the wild plants not only to infect indicator plants but also to infect tomato plants.  

Response 3: The reviewer is right; this would have been a necessary assay to show that in theory the ToBRFV that is in S. nigrum can infect back tomato. Actually, we did inoculation from S. nigrum infected plant to tomato plants, the tomato plants were positive by RT-PCR, however, a strong Tuta absoulta attack of the inoculated tomato plants prevented further evaluation of symptoms.

It is also important to point out, that not only a small segment was sequenced, but that we got the whole genome sequence of ToBRFV from symptomatic S. nigrum and the sequence turned to be almost the same (99.8% similarity) as sequence from tomato (not more different than different tomato isolates among themselves); then it is likely the same virus genetic pool that is shared in that environment.

We have added a sentence to address the importance of this point in the discussion.

Round 2

Reviewer 2 Report

I understand that the authors did not accept my suggestion to test the weeds as a source of tomato plants infection. However, since the authors emphasize the necessity of this test for future work and because the importance of their findings to the public I recommend to accept the corrected manuscript.

Author Response

The reviewer accepted the revised version without comments.